# Rationale, Feasibility, and Acceptability of the Meeting in Nature Together (MINT) Program: A Novel Nature-Based Social Intervention for Loneliness Reduction with Teen Parents and Their Peers

**DOI:** 10.3390/ijerph191711059

**Published:** 2022-09-04

**Authors:** Ashby Lavelle Sachs, Eva Coringrato, Nadav Sprague, Angela Turbyfill, Sarah Tillema, Jill Litt

**Affiliations:** 1Environmental Studies Program, University of Colorado Boulder, Boulder, CO 80309, USA; 2Barcelona Institute for Global Health (ISGlobal), Barcelona 08003, Spain; 3Department of Epidemiology, Mailman School of Public Health, Columbia University, New York, NY 10032, USA; 4Young Mother’s Clinic at the Children’s Hospital of Colorado, Aurora, CO 80045, USA

**Keywords:** loneliness, adolescent parent, nature-based, social prescription, community based participatory research, stress, green space, mindfulness, social connection

## Abstract

Recently, there has been an increase in feelings of loneliness and mental health conditions among adolescents. Within this population, parenting teens are at an increased risk for these conditions. Outdoor experiences are shown to be an antidote to loneliness and a way to promote social connectedness by amplifying the processes for supporting social relationships. In 2020–2021, we piloted the 8-week Meeting in Nature Together program (MINT) at a charter school for pregnant and parenting teenagers in Colorado, USA. MINT aimed to promote relatedness and nature connection for students ages 14 to 19. MINT included online and in-person group meetings with educational content, creative activities, discussion, park excursions, mindfulness activities, journaling, and nature photography. Here, we ask, can a school-level nature-based social intervention reduce loneliness among pregnant and parenting teens by promoting and sustaining social connections? How acceptable is MINT to participants? Methods included audiovisual recording transcriptions, surveys, and observation field notes. Results suggest that MINT fostered social connections through a tailored nature-based intervention delivered to a typically isolated community in culturally sensitive, developmentally appropriate ways. MINT proved feasible and effective as participants reported high levels of satisfaction and interest in continuing to engage in activities promoted in MINT.

## 1. Introduction

A large body of evidence suggests that feelings of loneliness are associated with an extensive array of adverse health outcomes (e.g., cardiovascular disease, inflammation, depression, anxiety, reduced mobility, and death) [1,2,3]. Studies have identified that loneliness is more prevalent in adolescence than in older populations [4]. Specifically, teen mothers have been identified for elevated risk of isolation and loneliness [4]. Becoming a parent of a newborn baby is stressful at any age as it comes with many new challenges (e.g., fatigue, juggling feeding and sleep routines) [5,6]. Additionally, teen mothers tend to have smaller networks of peers engaging in such life transitions, lower levels of education and income, and higher rates of single parenthood and physical abuse than adult mothers [7,8,9,10,11]. As such, adolescent and teen mothers face disproportionately high rates of loneliness [12,13,14].

The COVID-19 pandemic caused increased feelings of loneliness across the globe [15]. Prior to developing a COVID-19 vaccine, social distancing, quarantining, and isolation were the primary preventative efforts to reduce mortality and morbidity from the COVID-19 pandemic [16]. While these measures proved effective, these restrictions led to increases in self-reported loneliness worldwide [15,17]. Additionally, the COVID-19 pandemic has been identified as a threat-multiplying event [18]. As such, the COVID-19 pandemic exacerbated health disparities faced by marginalized groups [18]. Therefore, it is not surprising that the loneliness felt by adolescent mothers has increased over the past two years [12,19].

The COVID-19 pandemic also highlighted the importance of urban greenspace and nature on mental health and loneliness, as many people increased the amount of time spent outdoors in natural areas [20,21,22,23]. However, the increase in nature contact was not universal, as some marginalized communities saw reductions in time spent outdoors during the COVID-19 pandemic [18]. The increased disparities in nature contact during the COVID-19 pandemic proved to be problematic as a large body of evidence suggested that nature-contact may be a sustainable method for reducing the emotional and health-related burdens to loneliness [24,25]. Nature-based interventions are believed to support and nurture social connections, which consequently reduces the risk of loneliness [26,27]. Additionally, nature-contact has been shown to have an array of positive health benefits, such as reduction in cardiovascular disease [28,29], improved cognitive and brain development [30], reduction of depression [24,30,31], reduction of anxiety [24,32], increased quality of life [33,34,35], and reduced mortality [28,36]. As such, interaction with nature could potentially act as a buffer against the adverse health outcomes associated with loneliness [37].

Due to the growing body of evidence of the health promotive qualities of nature-contact, prescribing initiatives have become an increasingly popular intervention to promote health outcomes [27,38,39]. The concept of park prescriptions is gaining momentum, promoting park use and combining the benefits of increased physical activity with nature contact and exposure [39,40,41]. For example, in the USA, the National Park Service has partnered with the National ParkRx Initiative in San Francisco, CA with the goal of supporting the growing community of Park Prescription practitioners [42,43]. In addition, the Canadian PaRX program links health providers with resources and materials to prescribe nature experiences to promote a wide array of health benefits. PaRX is Canada’s first national nature prescription program [44]. Another novel nature prescription program that targets improved nutrition involves farmers’ market prescriptions, incorporating participant rebates for fresh produce [45,46].

Related to the concept of park prescriptions is the concept of social prescribing. Social prescribing originated in the UK and addresses people’s social connection needs holistically through activities that are typically provided by community organizations (e.g., arts activities, group learning, gardening) [47]. Nature-based, social prescribing (NBSP), also referred to as green social prescribing, is an innovative option to extend the reach of greenspaces such as parks and gardens to support health and wellbeing [27,48]. Care professionals can provide patients with nature-based social prescriptions by writing patients recommendations to spend time outside with others [27]. The majority of literature suggests that nature-prescribing programs are effective in making short-term behavioral change as well as helping to promote positive health outcomes [38]. However, the evidence base on NBSP is emerging, and more studies are required to measure their effectiveness [27]. Additionally, there is limited research on the efficacy of nature-based social prescriptions for isolated communities, such as adolescent mothers.

NBSP combines the need for nature contact with the need for social support. In this paper, we explore a NBSP intervention with adolescent participants via the Meeting in Nature Together program (MINT). Here, we present the MINT rationale, approach, and feasibility. The MINT program was an elective course for students at a charter school for pregnant and parenting adolescents during the fall 2020 and spring 2021 semesters. Community partners at the charter school and the Young Mother’s Clinic of the Children’s Hospital of Colorado (YMC) were concerned with loneliness among adolescent parents and their families. New solutions were needed beyond lists of resources, to ensure that young people who face loneliness are aligned with support systems to address these conditions. YMC and school staff were interested in nature-based activities and structured social support for lonely young patients. MINT is designed to support a community of adolescent parents in culturally sensitive, developmentally appropriate ways to amplify social connectedness. In this community-based participatory research study, we ask the following questions: Can a school-level nature-based social intervention reduce loneliness among pregnant and parenting teens by promoting and sustaining social connections? How acceptable is the program to our participants? How could MINT be improved in the future?

## 2. Materials and Methods

### 2.1. Study Setting

This MINT feasibility study took place at a charter school dedicated to the unique and complex needs of pregnant and parenting teenagers in Aurora, Colorado. The school provides critical services to young parents in the community, as safe and supportive spaces to care for their child or work towards their diploma. Services are provided in the form of quality education and parenting instruction, childcare, and personalized empowerment for parents and their children. Of the approximately 96 students at the school, 82% identify as female, and 18% as male. The student body is made of 2% Asian students, 76% Hispanic, 17% Black, 4% White, and 1% from two or more races [49]. Approximately 90% of the students are eligible for free lunch, and 2% are eligible for reduced lunch [49]. Not all students at the school are pregnant or parenting, as some relatives of young mothers choose to attend as well.

The City of Aurora is the most ethnically and racially diverse area in Colorado (Black/African American: 20%, Hispanic or Latino, 45%) [50]. Approximately 21% of residents in Aurora are born outside of the USA, and 33% speaking another language at home [51] There are numerous affordable ethnic grocers and markets supporting the established immigrant community in Aurora [52]. Aurora has many community assets to support young parent wellbeing and connectedness. Aurora boasts 8000 acres of open space, 97 parks, 3 nature centers, 6 recreation centers, and more [50]. These free and low-cost outdoor and indoor public spaces support social connection through family events and programming.

Aurora is adjacent to the community of Central Park (formerly Stapleton), where nearly 82% of residents are White, and the average annual wage is over $132,000 [53]. Central Park is close to the school and has many high-quality park assets. However, students and administrators described how students felt out of place and uncomfortable in Central Park. Another motivation for this study was to better understand neighborhood barriers inhibiting students from enjoying nearby park areas.

### 2.2. Intervention

MINT was designed to better understand the lived experience of local young parents as well as their childless fellow students at the school. We co-created an 8-week feasibility program through focused meetings with MINT community partners at the charter school and the YMC in the spring and summer of 2020. The program occurred during the school day for the fall 2020 and spring 2021 semesters. Due to the onset of the COVID-19 pandemic, we converted the in-person protocol to accommodate distance learning, and thus created an all-online intervention using Google Classroom, Google Meet, and self-guided, peer group-supported nature-based activities for the fall semester.

During both semesters, participants shared and listened to peers’ experiences with family, friends, the natural world, and community resources in the Aurora and Denver, Colorado areas. MINT was not a required course in the charter school’s curriculum. Study participants received financial incentives (in the form of bus passes and gift cards) and received elective school credits for their participation.

In fall 2020, the program met for one hour each week online, via Google meet. Activities began with team and trust building activities. As the sessions progressed, program components were adapted to promote environments and activities that supported and strengthened connectedness with this population and included open discussion, local park excursions, mindfulness activities, journaling, nature photography, and more.

In spring 2021, MINT shifted to a hybrid online/in-person model, adding in-person weekly park walks to the curriculum. We explored participants’ levels of loneliness pre and post MINT participation, as well as their satisfaction (acceptability) of MINT with a follow up survey at the end of the 8 weeks. Full ethics approval was granted by the University of Colorado, Boulder Institutional Review Board, protocol #20-0313.

### 2.3. Guiding Conceptual Frameworks for Intervention

The aim of the MINT curriculum is to minimize loneliness, increase self-value, and promote coping skills to balance school and parenting responsibilities. Ultimately, we intended to help build lasting communities where adolescent parents can engage with others in ways that support their values and goals. MINT recruited students from the charter school for pregnant and parenting teenagers in fall 2020 (September to November) and spring 2021 (February through May). MINT is a theory-informed intervention designed to support relatedness and social connectedness (Table 1). The curriculum was co-created by a team including social workers from the charter school, a clinical social worker from the Young Mother’s Clinic at the Children’s Hospital of Colorado, the director of the Population Mental Health & Wellbeing Program at Colorado School of Public Health, a PhD candidate in environmental studies with horticultural expertise, and a leading expert in urban environmental health, behavioral health, epidemiology, and community based participatory research.

### 2.4. Intervention Curriculum

MINT curriculum is summarized in Table 1. Data collection methods are outlined in Table 2. Sessions integrated theory and methods from appreciative inquiry [54], mindfulness practices [55,56,57,58], Self-determination theory [59], and Social cognitive theory [60]. Each session opened with a “Rose and Thorn” exercise where each participant described something positive and challenging from their week. Each session closed with a facilitator or student-led nature-based meditation. As the facilitators were primarily non-Hispanic White females and the student body was primarily Black and Hispanic (Table 3), guest speakers were intentionally selected that represented the participants’ demographics to provide a diverse representation of what it means to enjoy nature. Three weeks of additional nature walks with journaling activities were added at the end of the spring 2021 program, in response to student interest in continuing to meet after the official curriculum had concluded in Session 8. Attendance for each session is detailed in Table 4. MINT completion was measured by attaining 24 total points—three points were granted for attending each session, and bonus points were provided for submitting extra assignments, for example, completing nature challenges using the “Seek” by iNaturalist [61] mobile phone application.

### 2.5. Recruitment and Consent

We recruited participants after sharing information with students via Google Meet during a virtual school orientation week at the beginning of the fall 2020 and spring 2021 semesters. Interested students completed an online form, providing contact information for both student and parent (if they were under 18), preferred method of contact, and general schedule availability. Study staff contacted students to discuss their interest in the study and to answer questions. If the student wished to join, the project coordinator emailed the student’s parent a permission form in English and Spanish. If the student was over 18, this step was omitted. If permission was granted or if the student was an adult, the project coordinator emailed the student an assent form. High school teachers were not involved in recruitment to prevent perceived or actual coercion among students on campus.

### 2.6. Data Collection

For this paper, feasibility and acceptability were assessed by participant attendance, session transcripts, participant completion rates, and participant and administrator feedback. We used the University of California, Los Angeles (UCLA) Loneliness Scale, Version 3 to assess loneliness [63]. Demographic characteristics were self-reported. The post course survey included open-ended feedback fields with questions regarding how well participants connected with nature, how well they connected with others, if they felt valued and respected in MINT, and how the program could be improved. Administrator feedback was recorded via research field notes and text message communication to the first author. Data collection methods are outlined in Table 2.

### 2.7. Qualitative Analysis

The first and second author analyzed transcripts inductively without a predetermined codebook using ATLAS.ti software and Braun and Clarke’s thematic qualitative analytic method [62,64]. The second author reviewed a dataset subsample (15/23), approximately 65% of total transcripts. We coded all data iteratively, organizing codes into code groups and building visual network presentations to examine links between codes and concepts. We selected significant concepts by extracting repetitive topics in the data for closer analysis, sorting codes into larger code groups and visually networking these groups to reflect prominent themes [65]. We supported intercoder reliability via weekly online meetings where we discussed our code book application to MINT session transcripts, research field notes, session assignments, and facilitator feedback surveys [66,67].

### 2.8. Quantitative Analysis

Loneliness was assessed at baseline and post-intervention using the UCLA Loneliness Scale Version 3 [63]. The scale includes 20 items such as “How often do you feel that you lack companionship?” and “How often do you feel isolated from others?” rated from 1 (never) to 4 (always). An overall loneliness score was tabulated by summing all items in the scale producing scores ranging from 20 (low loneliness) to 80 (high loneliness). This scale has shown reliability and internal consistency in measuring loneliness among diverse populations including adolescents, young adults, and mothers [63,68,69,70,71]. Although the study was not adequately powered (given the small sample size of *n* = 8 in 2020 and *n* = 9 in 2021) to assess the impact of the intervention on loneliness scores, we were able to assess participants’ acceptance of the instrument and compute descriptive statistics using STATA version 13 (Texas, USA) for this feasibility study. These were different populations with slightly different interventions due to changing COVID-19 school policies, so the 2020 and 2021 cohorts are considered separately in the analyses.

## 3. Results

### 3.1. Participant Characteristics

Participant demographic characteristics are shown in Table 3. We recruited 8 students for the fall and 9 for the spring program, including both parenting and non-parenting teens who were friends or relatives of parenting teens, ages 14–19. Inclusion criteria included 9th to 12th grade students at the school. Most participants were female (75% in 2020 and 78% in 2021) and most were in their Junior or Senior year of high school (75% in 2020 and 77% in 2021). Within the 2020 cohort, 63% of participants identified as Hispanic or Latino, 13% identified as African American, and 25% indicated another race or ethnicity. Within the 2021 cohort, 50% of participants identified as Hispanic or Latino, 25% identified as African American, 8% identified as Caucasian or White, and 17% indicated another race or ethnicity. In both cohorts, all participants reported living with parents, guardians, or relatives and/or their children. In the 2020 cohort, the mean loneliness score was 47.38 (SD = 15.25) at baseline and 44.75 post intervention (SD = 10.70) and in the 2021 cohort, the mean loneliness score was 48.67 (SD = 7.071) at baseline and 50.00 post intervention (SD = 3.606).

### 3.2. Attendance and MINT Completion

In the fall, 75% of participants reached 24 points for official completion in fall 2020, and 33% in spring 2021.

### 3.3. Student Participant and School Administrator Feedback

Participant and administrator feedback is featured below in Table 5. Of the 17 total participants, 100% completed the feedback survey in fall 2021, and 44% in spring 2021. Of those who completed the survey (*n* = 12), 100% described feeling closer to nature, 83% felt connected with others in the program, and 100% felt respected and valued in MINT. The results were categorized into 5 major themes in the qualitative coding process described above in Section 2.8.

#### 3.3.1. Mindfulness, Mental Wellness Aspects Resonated with Participants

Feedback characterized in this category incorporated responses from the feedback survey, communication from a school social worker, and comments by participants during MINT sessions. Results reflected participant appreciation for the stress relief they experienced, with students mentioning the meditation helping them to feel calm or being outdoors helping them feel more relaxed and more like themselves. By the end of each semester program, several participants (*n* = 3) began leading the closing meditation sessions for their fellow students.

#### 3.3.2. Participants Enjoyed Engaging with Nature Photography Elements

This category includes comments by students on feedback surveys, when asked how the program could be improved. Several participants (*n* = 4) mentioned the mobile phone nature photography assignments as their favorite aspects of the program, as it motivated them to get outside more often. In addition, the photography served as a vehicle for more in-depth photo elicitation sessions in which students shared their hopes and preferences for spending time in nature, and barriers keeping them from spending time outdoors. This creative aspect helped motivate more sharing between participants and provided structure, as nature photography was a core aspect of the curriculum. One adjustment to meeting in person in spring 2021, was that it was more difficult to review participant photography together outdoors than it had been online. Unsurprisingly, students preferred to walk and talk with one another, rather than attempt to review photography outdoors on an iPad. We printed and framed student photographs for an in-person end of MINT celebration and photography exhibition at the school in spring 2021.

#### 3.3.3. MINT Provided the Opportunity for Participants to Connect and Feel a Sense of Belonging

More than half of students described enjoying the supportive sense of community that MINT provided. Participants expressed how they felt more sociable and connected to each other as MINT progressed, and feeling cared for, and feeling part of a group. The elements of the program cited by students that encouraged this were sometimes connected to the facilitators’ capacity to encourage group sharing. Moreover, the feedback included mentions of how much the participants enjoyed meeting one another. When meeting in person, researchers observed that due to the COVID-19 pandemic online schooling, many students did not actually know many others at the school outside of MINT. MINT helped to fulfill the basic need of relating with others and feeling belonging at school.

#### 3.3.4. MINT Motivated Participants to Deepen Their Relationship to the Natural World

Almost all students shared either in the survey or during sessions how the program helped them connect with nature more deeply. Many already had a keen interest in the outdoors. Program elements such as nature photography, or homework assignments to go for a walk and see what students noticed were cited in surveys and in sessions as motivating to spend more time outdoors. Students described enjoying seeing how others appreciated nature, realizing how important nature was to them, or how it made them more relaxed, and happier.

#### 3.3.5. Participants Needed More Emotional Support during the COVID-19 Pandemic

This theme encompasses critical feedback received by participants in the feedback survey. Participants (*n* = 2) suggested meeting more in person, and one specifically recommended having more in-depth check-ins with students.

### 3.4. MINT Feasibility

MINT proved feasible and effective as participants reported high levels of satisfaction and interest in continuing to engage in activities promoted in MINT. In the fall 2020 program, acceptability of the intervention was high, based on attendance rates, course completion rates, pre-and post-loneliness scores, and feedback surveys. An average of 83% of participants attended each session, 75% achieved the necessary number of participation points to complete the course and receive school credit, pre and post loneliness scores were lower following MINT. Of total participants who completed follow-up surveys (*n* = 12), 100% positively assessed the acceptability of the program.

## 4. Discussion

This preliminary study explored the feasibility of an 8-week nature-based social intervention for adolescent parents and their peers. The intervention, which was delivered during the school day at a charter school for pregnant and parenting teens in Aurora, Colorado, aimed to increase social connectedness through engaging in socially supported nature activities. We co-created the program in collaboration with our community partners to tailor it to our target population’s needs and lifestyles. We used participatory processes throughout the program. Our objective in designing MINT was to understand the lived experience of adolescent parents and to evaluate the efficacy of nature contact to teach and promote coping skills and social connection. We sought to determine if our school-level nature-based social intervention served as an effective way to address loneliness among pregnant and parenting teens, how acceptable was MINT to our participants, and how could MINT be improved.

### 4.1. COVID-19 Pandemic Environment

One year into the COVID-19 pandemic in early 2021, MINT researchers struggled both in recruiting spring 2021 participants and motivating MINT attendance. Here, we discuss potential reasons for these differences between semesters as well as solutions moving forward. It was clear following conversations with school administrators in spring 2021, that school attendance was decreasing one year into the pandemic. These challenges reflected a nationally representative teacher survey taken during the pandemic in the USA showing that 23% of students had no contact with teachers and an additional 45% of students had much lower levels of engagement than before the pandemic [72]. In addition, the COVID-19 pandemic isolation increased levels of loneliness globally [15,17], particularly exacerbating adolescent loneliness’ and mental health concerns [12,19]. In this study, we found that students were more motivated to meet and engage online in fall 2020 after months of pandemic related isolation. By spring 2021, student interest in meeting online had declined, and mental health concerns were worsening after a full year of pandemic-related restrictions. We theorize that this atmosphere provides context to spring 2021 student attendance and course completion challenges and loneliness rates observed in 2021. As online participation waned, we received parental and ethical permission to meet in person for weekly nature walks in spring 2021. During these walks, students shared about the challenges of pandemic isolation.

### 4.2. Participant Loneliness Levels

Loneliness was captured both qualitatively and quantitatively. Baseline loneliness scores from all MINT participants were captured successfully. Follow up was difficult in spring 2021 because overall attendance was low. Baseline loneliness scores from MINT participants indicated moderate loneliness both pre and post intervention [73]. Loneliness scores from MINT participants were similar to those measured among international samples of adolescents and young adults during the COVID-19 pandemic [74,75,76,77,78], indicating congruence between the sample in this study and other populations of adolescents. Given that the loneliness scores obtained in this study were measured during the COVID-19 pandemic, the scores may not be indicative of typical levels loneliness within this population. It may instead indicate heightened loneliness due to pandemic isolation. The qualitative analysis described below underscores how the program provided a sense of belonging to numerous participants, thereby increasing their sense of connectedness, and potentially reducing feelings of loneliness.

### 4.3. Participant Sense of Belonging

The theme of experiencing belonging in MINT was the most coded theme in our dataset. Feedback suggests that the socially connecting nature of MINT was most appreciated, and most needed by students during this challenging time for teens. Almost all participants in both cohorts shared feedback describing how they enjoyed connecting with their fellow participants. They described how they felt calmer and more connected to themselves and to the natural world. These positive responses reflect research showing that connecting with others in nature can break down barriers between community members, increase feelings of connectedness with others and reduce stress [27,79,80]. Connecting with nature in MINT was the second most coded theme in our qualitative dataset. MINT findings build upon research suggesting that outdoor experiences may facilitate social involvement and bonding [81]. Research with young mothers shows that social support may increase their confidence in caring for their children, and positively influence their transition to motherhood [82]. In addition, socially supported adolescent mothers are five times less likely to develop post-partum depression [83]. Connecting adolescent parents with comprehensive services to meet their social, health and educational needs could potentially improve long-term outcomes for both parent and child [84].

### 4.4. Limitations

Launching MINT at the beginning of the COVID-19 pandemic introduced both logistical and scientific challenges. Pandemic conditions complicated participant recruitment and retention, as students were weary of online learning by 2021 and reluctant to register for the program and attend sessions online. Lower attendance in spring 2021 made it difficult to evaluate MINT hybrid aspects when we began both meeting in person and online. In addition, the high-risk nature of the population made implementing new programming a challenge, as adolescent parents understandably had little free time and difficulties at home interfered with coming to class. We were unable to do significance testing in this feasibility study, due to the small sample size. In addition, we did not collect data on gender diversity in this study. In future MINT studies we will, as gender identity and expression may influence adolescent acceptability of and adherence to health promotion programs.

### 4.5. Future Directions

Future research using this model could assess if MINT could be expanded as the basis of a nature-based social prescribing program adopted by clinics and social services organizations. We aim to use this feasibility study as a guide for improvement. In future MINT studies, we aim to increase the sample size and add a control group to test this model’s efficacy as a social connectedness intervention. Furthermore, we find that the facilitator-led model and focusing on community building within the cohort was critical to engaging youth and sustaining participation. For study retention, we aim to increase in person programming to meet the need for more social support. However, we also observed that virtual meetings accommodate young mothers’ limited schedules, so using a hybrid online and in person model is acceptable with this population.

## 5. Conclusions

Our objective in designing MINT was to understand the experience of young adults via a network of adolescents raising children and their peers, to understand if MINT served as an effective way to address loneliness in this population and how acceptable it was to participants. We followed a co-creation process to best tailor the program to our target population and used participatory processes throughout to better understand participants’ lived experiences. We analyzed adolescent parents and their peers’ experience following the 8-week school-based feasibility intervention designed to increase social connectedness through nature connection with others. We find that MINT served as a feasible and acceptable nature-based social intervention and that it was delivered to a typically isolated community in culturally sensitive, developmentally appropriate ways. MINT has strong potential to be part of a NBSP approach. MINT participants reported high levels of satisfaction and interest in continuing to engage in activities promoted in MINT.

## Figures and Tables

**Table 1 ijerph-19-11059-t001:** MINT Curriculum.

Timing and Theme	Session 1: Fall 2020	Session 2: Spring 2021	Theoretical Rationale
**Session 1: Connecting Counts**	**Online:** General MINT overview, creating shared agreements about participation and establishing an open and accepting atmosphere. The group reviews some nature quotes together and discusses a time when they each felt connected to nature and others.		Self-determination theory, mindfulness practices
**Session 2: Stepping out**	**Online:** Social network mapping exercise, students draw a connected map of their social relationships. The group shares and discuss social worlds, and what areas of life are the most and least social.		Appreciative inquiry, Self-determination theory, mindfulness practices
**Session 3: Surrounding ourselves**	**Online:** Guest speaker leads a discussion on positive relationship skills. Students draw on lesson plans to list as a group how to surround themselves with supportive people. The students discuss a list of positive relationship qualities that help build a strong social network.		Appreciative inquiry, Self-determination theory, Social Cognitive Theory, mindfulness practices
**Session 4: Nature photography**	**Online:** Discuss examples and strategies for nature photography. The facilitator demonstrates how to use the iNaturalist mobile phone application [62] to track plant and animal species nearby.	The same activities, **in person** in local park, with a nature walk	Self-determination theory, mindfulness practices
**Session 5: Creating collectively**	**Online:** Guest speaker: a local intuitive medicine practitioner and former teen mother led a tea ceremony after mailing packets of tea to each student. The facilitator also selects student photography to review as a group.	**In person** in local park: a local artist/naturalist DACA (Deferred Action for Childhood Arrival) recipient from the neighborhood shared his nature photography, cameras, and nature mandalas made with invasive species. (DACA legislation supports young people who entered the USA unlawfully as children).	Self-determination theory, Social Cognitive Theory, mindfulness practices
**Session 6: Making time**	**Online:** Participants each draw a 2-week calendar in their journals. The group discusses time management and making time for social connection between family and school. The students manually block out time in the calendars for social activity, school, and family for a sample week. The group then discussed their typical weeks and challenges they faced in connecting with others due to parenting responsibilities. The facilitator also selects student photography to review as a group.		Appreciative inquiry, Self-determination theory, mindfulness practices
**Session 7: Nature is for everyone**	**Online:** Guest speaker, Executive Director of a Latinx outdoor leadership nonprofit leads discussion on his work and diversity in the outdoors. The facilitator also selects student photography to review as a group.		Self-determination theory, Social Cognitive Theory, mindfulness practices
**Session 8: Ideal Day**	**Online:** Participants spend a few moments journaling in the beginning of class. The prompt focuses on their idea of an ideal day outdoors. The group shares what their ideal day outdoors would look like. Select additional student photographs to review as a group. Close by summarizing key themes from the nature photography, short nature meditation.	The same activities, **in person** in local park, with a nature walk	Self-determination theory, mindfulness practices
**Session 9: Park walk**	Only in spring session	**In person** at local park, with group check-ins, nature walk, journaling, and meditation	Self-determination theory, mindfulness practices
**Session 10: Park walk**	Only in spring session	**In person** at local park, with group check-ins, nature walk, journaling, and meditation	Self-determination theory, mindfulness practices
**Session 11: Park walk**	Only in spring session	**In person** at local park, with group check-ins, nature walk, journaling, and meditation	Self-determination theory, mindfulness practices

**Table 2 ijerph-19-11059-t002:** Data Collection Methods.

Method	Description
**Audiovisual recording**	Each Google Classroom session was recorded and transcribed by study staff.
**Participant questionnaire**	We collected demographic data and baseline loneliness data [63] on study participants with a pilot questionnaire. After the program ended, participants filled out a feedback form on their final reflections from the study and another loneliness survey.
**Participant observation**	After each session, Author 1 wrote field notes of her impressions of participant conversations and actions.
**Facilitator evaluation**	Author 1 filled out a facilitator evaluation after each session to reflect on the fidelity, dose delivered, and reach of the session aligned to MINT objectives.
**Session attendance**	Participant attendance in each session was documented. Students received incentives in the form of gift cards and bus passes for attending and completing the program.

**Table 3 ijerph-19-11059-t003:** MINT 2020 and 2021 Cohort Demographics and Loneliness Scores.

Variable	2020 Cohort (*n* = 8)		2021 Cohort (*n* = 9)	
Demographic Characteristics	Mean (Range) or *n* (%)		Mean (Range) or *n* (%)	
Age	17.35 (14–18)		17.52 (15–19)	
Sex				
Female	6 (75%)		7 (78%)	
Male	2 (25%)		2 (22%)	
Education				
Freshman	1 (13%)		1 (11%)	
Sophomore	1 (13%)		1 (11%)	
Junior	0		3 (33%)	
Senior	6 (75%)		4 (44%)	
	Frequency (% of responses)		Frequency (% of responses)	
Race/Ethnicity ^3^				
Caucasian/White	0		1 (8%)	
African American/Black	1 (13%)		3 (25%)	
Asian/Asian-American	0		0	
Hispanic/Latino	5 (63%)		6 (50%)	
Native American	0		0	
Other	2 (25%) ^1^		2 (17%) ^2^	
Living Situation ^3^				
Alone	0		0	
With other students	0		0	
With roommates who are not students	0		0	
With parent(s), relative(s), or guardian(s)	8 (80%)		8 (80%)	
With a husband/wife domestic partner/significant other	0		0	
With my child/children	2 (20%)		2 (20%)	
**Outcomes**	**Completion** ***n* (%)**	**Mean (SD)**	**Completion** ***n* (%) ^4^**	**Mean (SD)**
Loneliness				
Initial	8 (100%)	47.38 (15.25)	9 (100%)	48.67 (7.07)
Follow Up	8 (100%)	44.75 (10.70)	3 (33%)	50.00 (3.61)

^1^ Other race/ethnicity included multiracial and Pacific Islander. ^2^ Other race/ethnicity indicates race was not specified. ^3^ The totals in these categories are higher than the total sample size, because participants had the option to select multiple options for both Race/Ethnicity and Living Situation. ^4^ In the 2021 cohort, all participants completed a baseline loneliness questionnaire, while only 3 participants completed the loneliness questionnaire post-intervention due to participant absenteeism.

**Table 4 ijerph-19-11059-t004:** MINT Session Attendance.

Session	Attendance Fall 2020	Attendance Spring 2021
**1**	8 (100%)	4 (44%)
**2**	5 (63%)	8 (89%)
**3**	5 (63%)	4 (44%)
**4**	6 (75%)	3 (33%)
**5**	8 (100%)	3 (33%)
**6**	7 (88%)	4 (44%)
**7**	6 (75%)	5 (56%)
**8**	8 (100%)	4 (44%)
**9 ***	n/a	4 (44%)
**10 ***	n/a	3 (33%)
**11 ***	n/a	3 (33%)

* in-person nature walks added in spring 2021 by group request.

**Table 5 ijerph-19-11059-t005:** MINT Participant and Administrator Feedback.

Key Finding	Examples of Participant/Administrator Quotation	Respondent
**Mindfulness, mental wellness aspects resonated with participants**	“I loved how [the facilitator] would share pictures with everyone and meditated, it helped me feel really calm”	Female, age 18, fall 2020
	“I am meeting with [student] right now and she is super close to graduating this year but needs another elective credit. She was talking about how much she got out of your class and how helpful it was for her mental health. Would she be able to take the class again? Or be a teacher’s assistant? I understand if neither are an option, we just thought we would ask!”	Female, school social worker
	“I feel like good and like stress relief as soon as I go outside, and I feel more active when I come inside…Yeah, like more creative, and it makes me just feel so good about myself”	Male, age 18, Fall 2020
**Participants enjoyed engaging with nature photography elements**	“I liked the requirement of having to take pictures every week because it motivated me to keep going outside getting air even for just a second.”	Female, age 19, Fall 2020
	“I would recommend that you keep where students have to take pictures of nature and upload them. That inspires me to go take more pictures outside.”	Male, age 18, Fall 2020
	“I would recommend that you guys keep the photograph aspect of the program because that was my favorite part of the program”	Female, age 17, spring 2021
**MINT provided the opportunity for participants to connect and feel a sense of belonging**	“...I never thought that I would feel so good with people I barely knew.”	Female, age 16, fall 2020
	“The way I would describe my sense of connecting with others during the program was a right way, I loved every one that was in the program they were nice and supportive.”	Male, age 17, fall 2020
	“…i felt like i was being cared for and part of the group”	Female, age 19, spring 2021
	“During the program I was comfortable with connecting with people. It was the right balance of getting to know/connect with people and the right balance of how I’m comfortable with nature”	Female, age 17, spring 2021
	“I started to be more social and comfortable with others because [the facilitator] made us feel like we could talk and communicate with each other”	Female, age 19, fall 2020
	“This experience was amazing I loved meeting everyone in the group”	Male, age 17, spring 2021
**MINT motivated participants to deepen their relationship to the natural world**	“... I saw [MINT] as an opportunity to connect to nature as I was when I was a child. During the program I saw how different people connect with nature and what nature means for them. In the beginning it was hard for me to connect with nature but as the program/time went by nature became more in depth into my subconscious, sometimes I would get thoughts out of nowhere during my day of going in nature and disconnecting myself from my phone. The program has helped me take the first step, which are always the hardest, to help me connect with nature.”	Female, age 17, spring 2021
	“My experience in connecting more with nature... I feel more relaxed and happier.”	Female, age 16, fall 2020 student
	“I loved it. It makes me realize more about nature and how important it is”	Female, age 17, fall 2020
	“I thought you would like to hear some fun news. [Student]’s family came and told me how much he is getting out of your class and how he loves it. They said he used to never get outside and how he is taking the family for walks! His cousin I believe [student] wants to join the class too so they can take walks together and get outside more.”	Female, school social worker
**Participants needed more emotional support during the COVID-19 pandemic**	“I feel like meeting more in person and really checking up on how everyone is doing [is needed]”	Male, age 17, spring 2021
	“[I suggest to] Meet more just to know each other and talk”	Female, age 16, spring 2021

## Data Availability

The data presented in this study are available on request from the corresponding author (J.L.). The data are not publicly available due to privacy concerns for research participants.

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
