# Peer review of "Rationale, Feasibility, and Acceptability of the Meeting in Nature Together (MINT) Program: A Novel Nature-Based Social Intervention for Loneliness Reduction with Teen Parents and Their Peers"

_ijerph, 2022, doi:10.3390/ijerph191711059_

Round 1

Reviewer 1 Report

The article entitled “Rationale, feasibility, and acceptability of the Meeting in Nature Together (MINT) program: A novel nature-based social intervention for loneliness reduction with teen parents and their peers” presents the Meeting in Nature Together (MINT) program, which Its purpose is to promote the relationship and connection with nature in adolescents and their families. The topic and the article are interesting. In fact, there are few papers in the literature on intervention programs and health promotion in adolescents, therefore, this article is innovative. Now, there are some things that I think should be checked before publication. Which I list below.

-In the method, the authors must dedicate a space to explain the characteristics of the people who participated in the program (participants), how many responded to the first call? How many finally participated in the program? What ages were they? … That information is found in the results, why? I recommend introducing the participants to the method.

-The authors make an effort to present the ethnic diversity of their participants, I understand that this information may be relevant when discussing the results. However, it is curious to me that they do not do the same with gender diversity. Different studies show that gender identity, sexual orientation and gender expression can influence adherence to health promotion programs in adolescents. If the authors did not take this diversity into account, they should state it as a limitation.

-The authors do not present statistical analyzes showing the effectiveness of the program on loneliness.

In general, it seems to me an interesting article, with many successes with your program.

Author Response

Thank you for your careful review and comments. Please see our responses below. 

Comment #

Section

Reviewer’s comment

Author’s Response

1

Results

MINT participant characteristics are recommended to move from Results to Methods section

Thank you for this suggestion. We agree that a dedicated space is essential for describing the study participants. We feel that participant characteristics are appropriately placed in this paper as a result. The method section describes the setting and recruitment methods from which we recruited our participants. The results section shows who agreed to participate in the study. We have added participant age ranges to this section.

2

Results

Participant gender diversity and expression is missing from the paper

We can see that collecting data on gender diversity and expression could have been an important metric in this study. We therefore added this language to the limitations section, and will collect this datapoint in future MINT studies:

In addition, we did not collect data on gender diversity in this study. In future MINT studies we will, as gender identity and expression may influence adolescent acceptability of and adherence to health promotion programs.

3

Results

Authors do not present statistical analyzes showing the effectiveness of the program on loneliness

We appreciate this suggestion. We debated on how to present the pre-post survey data. What we have shown in table 3, listing means and SDs, seemed the most valid way to discuss the scores in this small sample. As this is a feasibility study, the statistics are descriptive given the small sample size and low follow up response rate in the 2021 cohort (n=8 for 2020 cohort and n=9 for 2021 cohort). These were different populations with slightly different interventions, so the 2020 and 2021 cohorts needed to be considered separately for the analyses. There is little information on loneliness scores for this population in general, so our descriptive results are still useful and contribute to the literature.

We have updated section 2.8, quantitative analysis, below:

Although the study was not adequately powered (given the small sample size of n=8 in 2020 and n=9 in 2021) to assess the impact of the intervention on loneliness scores, we were able to assess participants’ acceptance of the instrument and compute descriptive statistics using STATA version 13 (Texas, USA) for this feasibility study. These were different populations with slightly different interventions due to changing COVID-19 school policies, so the 2020 and 2021 cohorts are considered separately in the analyses.

Reviewer 2 Report

The paper titled Rationale, feasibility, and acceptability of the Meeting in Nature Together (MINT) program: A novel nature-based social intervention for loneliness reduction with teen parents and their peers is on target.

Despite the effects that the Covid-19 Pandemic produced on the development of the programs, it was adapted to the circumstances, making it easier for adolescents to participate through the proposed hybrid sessions.

Obviously, studying society during a stage in which it required being isolated is very difficult to carry out, but training or contact through audiovisual media made that "loneliness" more bearable.

It would be interesting to carry out this program today, to see if the effects with direct intervention are possible in the same population, and to see the level of social support and group ties that have been generated.

Also, as a future proposal, apply the MINT program in another population, since as they state, the effects of Covid-19 have generated a serious mental health situation in the general population and in all sectors, to take into account other study variables : cities, ages and/or professions.

Author Response

Thank you for your careful review of our manuscript. We look forward to extending the reach of this program in other contexts and also continuing to work with our community partners to test this approach under different circumstances post-COVID.